# Heartbeat Detection by Laser Doppler Vibrometry and Machine Learning

**DOI:** 10.3390/s20185362

**Published:** 2020-09-18

**Authors:** Luca Antognoli, Sara Moccia, Lucia Migliorelli, Sara Casaccia, Lorenzo Scalise, Emanuele Frontoni

**Affiliations:** 1Department of Industrial Engineering and Mathematical Sciences, Università Politecnica delle Marche, 60121 Ancona, Italy; l.antognoli@staff.univpm.it (L.A.); s.casaccia@staff.univpm.it (S.C.); l.scalise@univpm.it (L.S.); 2Department of Information Engineering, Università Politecnica delle Marche, 60121 Ancona, Italy; l.migliorelli@pm.univpm.it (L.M.); e.frontoni@univpm.it (E.F.); 3Department of Advanced Robotics, Istituto Italiano di Tecnologia, 16163 Genoa, Italy

**Keywords:** laser doppler vibrometry, machine learning, support vector machines, contactless measurements, heartbeat, heart rate detection

## Abstract

*Background:* Heartbeat detection is a crucial step in several clinical fields. Laser Doppler Vibrometer (LDV) is a promising non-contact measurement for heartbeat detection. The aim of this work is to assess whether machine learning can be used for detecting heartbeat from the carotid LDV signal. *Methods:* The performances of Support Vector Machine (SVM), Decision Tree (DT), Random Forest (RF) and K-Nearest Neighbor (KNN) were compared using the leave-one-subject-out cross-validation as the testing protocol in an LDV dataset collected from 28 subjects. The classification was conducted on LDV signal windows, which were labeled as *beat*, if containing a beat, or *no-beat*, otherwise. The labeling procedure was performed using electrocardiography as the gold standard. *Results*: For the *beat* class, the f1-score (f1) values were 0.93, 0.93, 0.95, 0.96 for RF, DT, KNN and SVM, respectively. No statistical differences were found between the classifiers. When testing the SVM on the full-length (10 min long) LDV signals, to simulate a real-world application, we achieved a median macro-f1 of 0.76. *Conclusions:* Using machine learning for heartbeat detection from carotid LDV signals showed encouraging results, representing a promising step in the field of contactless cardiovascular signal analysis.

## 1. Introduction

Monitoring cardiac activity is of valuable importance in clinical applications as well as in non-clinical environments. The cardiac activity can be related to different factors, such as gender, age, physical and physiological condition [1]. Cardiac evaluation is crucial for the diagnosis and prognosis of several pathologies, such as myocardial infarction and neuropathy [2]. Heart rate assessment is of major importance for cardiac evaluation. Conventional methods for the evaluation of heart rate are electrocardiography (ECG) and photoplethysmography (PPG). Both are contact methods. The ECG uses electrodes to measure electric variation due to myocardial polarization. PPG is an optical method that detects the amount of transmitted or reflected light from the peripheral vessels to measure the pulsatile oxygen saturation [3]. PPG is less accurate when compared to the ECG because PPG may be affected by a motion artifact. Furthermore, PPG is less reliable for longer term recordings. Contact devices, such as ECG, may present issues. ECG may induce infection hazards caused by unsterilized lead wires [4]. Another issue is represented by performing ECG-based monitoring during magnetic-resonance (MR) scanning [5]. Safety guidelines require that all wires should be positioned straight inside the MR scanner to avoid the induction of currents by the MR magnet. However, the risk of severe burns still remains. Moreover, the demand for ubiquitous measuring of human physiological parameters is increasing not only in clinical fields but also in sport applications, home health care, etc. For this reason, contact devices may be annoying or uneasy for continuous measurement.

Non-contact cardiac monitoring systems are used to record continuously individual’s vital signs in a non-intrusive way. Those kinds of systems commonly find applications in heartbeat monitoring. In [6,7], a technology based on doppler radar is presented to non-contact heartbeat monitoring. This technology relies on the reflection properties of a body irradiated by high frequency radiowaves. The amplitude of the backscattered signal correlates with the internal and external tissue movement. With this technique, it is possible to penetrate some non-metal media, such as wood, clothes and water, and remotely sense human physiologic parameters. The use of high frequencies allow achieving high displacement resolution. Despite these properties, this method is affected by movement artifacts and presents high variability between subjects (in terms of different physical constitutions, breathing, size and position of the heart) [7]. In [8,9], heartbeat is measured by means of spectral analysis performed on the pulsatile photoplethysmographic signal from video recordings of subjects’ face. This technology allows to measure the heartbeat from multiple subjects (i.e., all subjects in the camera field of view) but the ambient light must be considered a noise source. In [10,11], thermal images are used for heartbeat detection. The method is based on the principle that the skin temperature in proximity of major superficial arteries depends on the pulse blood flow. However, the signal to noise ratio is low and thermal distortion (due to sweating, air flow and heat radiation) represents a source of measurement noise difficult to attenuate. As a result, environmental condition control is required. In [12], a method based on speech signal analysis is presented. Hence, the vocal cord vibration frequency is dependent on the heart activity. The method allows to estimate the heart rate using frequency modulation of the speech signal within a certain frequency band.

A promising contactless technique is Laser Doppler Vibrometry (LDV). LDV uses a laser beam and exploits an interferometric optical scheme to detect the velocity, along the direction of the optical beam, of a point on a surface [13]. Originally, LDV was mainly used for testing mechanical vibration in structures (e.g., wind turbine [14], washing machine [15], structural analysis [16], propeller [17]) or detecting defects in construction while avoiding direct contact with surfaces [18]. More recently, LDV has found application when a non-contact approach is required for evaluating cardiovascular function [19,20]. The interest in the use of LDV lies in the fact that, being a non-contact measurement, it does not interfere with vascular dynamics. Moreover, it is not affected by metal and radio reflective objects unlike electromagnetic sensors. LDV is also immune to environmental light conditions, which could, instead, compromise methods based on video analysis. By pointing the LDV laser beam on the neck, over the carotid (as shown in Figure 1), the acquired LDV signal can be used to measure the common carotid pulse [21,22,23]. Hence, the LDV detects, in a contactless way, vessel distention due to blood flow. This distension indirectly allows to retrieve systolic pulse and vascular dynamics. The choice of the carotid is due to the carotid proximity with respect to the heart [24].

Most of the studies in the literature relevant to LDV signals are devoted to understanding how to define proper measurement protocols for LDV signal acquisition. In [25], to understand the influence of the location of the LDV measurement spot, multiple LDV acquisitions, in correspondence to different locations on the neck, are analyzed. The work shows that the systolic peak on the LDV signal changes according to the measurement spot location. The maximum peak amplitude is found in proximity of the carotid, detected by palpation. In [26], a study is performed to understand how the blood passage in the jugular may affect the LDV measurement related to the carotid artery. To this goal, an ultrasonic doppler is used. In [27], thorax LDV signal acquisition is studied using reflective tapes. The results achieved suggest that using the reflective tape increases the LDV signal quality. Nonetheless, this may pose issues because the tape has to be attached directly on the subject’s skin.

State-of-the-art approaches to LDV signal analysis are mainly based on visual analysis (e.g., [28]). Other work implements frequency-based analysis coupled with filtering techniques. In the work proposed in [19], LDV peaks are selected as the first local maximum values, in the LDV signal, that follow the R-peak in the ECG signal. The tachograms of ECG and LDV signals are obtained and power spectral analysis is performed to measure the ratio between the low and high frequency, as an estimation of the sympatho-vagal balance. In [22], LDV signals from the carotid and chest are recorded simultaneously and compared with ECG. The Pan–Tompkins algorithm is used to detect the R peak in the ECG, and the semi-automatic methodology proposed in [19] is implemented to find the systolic peak in LDV carotid signal. Heart periods are calculated from ECG and LDV, and tachograms are obtained. Spectral analysis of the tachograms is performed to estimate the sympatho-vagal balance.

In [24], carotid LDV and ECG are acquired simultaneously. Heart periods are computed from the R peaks of the ECG and the corresponding peaks in the LDV, which are identified by means of a semi-automatic algorithm based on [29]. Other researchers (e.g., [23,30,31,32,33]) focus on estimating the pulse-wave velocity, which measures the speed of propagation of pulse waves along the arterial tree. In this work, the analysis still relies on manual intervention and/or ECG signal availability.

To tackle the LDV variability and to perform an automatic analysis, machine learning (ML) approaches may be used. Such variability in LDV signals arises from several factors, including heart-rate (HR), breathing patterns, and mental and physical stress [34,35,36]. ML has already been proved to be successful for several tasks relevant to ECG signal analysis [37,38]. Nonetheless, no efforts have been put in LDV signal analysis, so far. ML classifiers rely on a training procedure, which may be costly in terms of computational resources and time. However, today this does not represent an issue, considering that GPU resources are available also for free for research purposes (e.g., Google Colab (https://colab.research.google.com/)). Considering the potentiality of ML for signal analysis in closer fields, and with a view to tackle LDV signal variability, the aim of this work is to assess whether ML classifiers can be used for detecting a heartbeat from the LDV signal.

With such a goal, we collected a dataset with LDV signals acquired from 28 subjects. Relying on this dataset, we designed an experimental study consisting of a leave-one-subject-out cross-validation strategy.

Thus, guided by the research hypothesis that ML models can detect heartbeats from LDV signal acquired from the carotid artery, the main contributions of this paper are:A novel method for the assessment of the carotid heartbeat from LDV signals, using an ML framework for heartbeat detection (Section 2.3).A comprehensive validation on real LDV signals (Section 2.3): validation on the LDV signals acquired from a specifically created dataset of 28 subjects to experimentally investigate the research hypothesis.The release of the LDV-beat dataset (Section 2.2): collection of the first and largest dataset (the LDV-beat dataset) of carotid LDV signals from 28 subjects (for a total of 280 min of recording) with corresponding ECG signals, used as gold standard for beat detection. The dataset, will hopefully foster ML research in the field of cardiac signal processing with LDV.

To the best of our knowledge, this is the first attempt to investigate ML approaches for automatic beat detection from the carotid LDV signal. For promoting research in the field, our dataset is fully available online (https://zenodo.org/record/3892521).

## 2. Material and Methods

The workflow of the proposed framework for heartbeat detection from LDV-signal is shown in Figure 2.

The proposed method consists of three main steps: LDV-signal acquisition (Section 2.1), LDV-beat dataset creation (Section 2.2) and LDV-signal classification (Section 2.3).

### 2.1. LDV-Signal Acquisition

The measurement setup used to collect the LDV-beat dataset is shown in Figure 1. The setup included a class-2 (eye-safe) laser vibrometer (model PDV-100, Polytec, Germany) (www.polytec.com), an acquisition board (PowerLab 4/25T, ADInstrument, UK)(www.adinstruments.com/products/powerlab) and a laptop PC to record the data. The LDV system used a Helium Neon (HeNe) laser source with a wavelength of 633 nm and a vibrational velocity resolution of 0.02 μm/s. The output of the device was an analog voltage with a conversion factor of 5 mm/s/V.

The LDV signals were recorded from 29 healthy subjects (7 males and 22 females) aged between 21 and 25 years. All participants were informed and provided a written agreement in accordance with the Declaration of Helsinki.

No specific treatment was performed on the subject’s skin prior to performing LDV signal acquisition. The measurement distance between the laser optical head and the subject was about 0.7 m. The LDV laser was pointed on the subject neck at about 1 to 2 cm below the right carotid sinus, in correspondence of the carotid artery wall. The carotid position was detected manually by palpation. The subjects involved in the study were lying in a supine position to limit, as much as possible, involuntary movements that could affect the LDV-signal acquisition. Each acquisition lasted 10 min. One subject was excluded because the LDV signal has a low signal to noise ratio. This happened because the laser was not pointed properly over the carotid artery.

In this work, the ECG was used as the gold standard for heartbeat detection. Thus, an ECG medical device (Adinstrument PowerLab 90 4/25T, UK) was used to acquire II-lead signals. The LDV and ECG signals were recorded synchronously with a 24 bit A/D converter and a sampling frequency of 500z Hz.

### 2.2. LDV-Beat Dataset Creation with a Windowing Approach

A sliding-window algorithm was used to process the LDV signal to detect heartbeat. The LDV signal was divided in overlapping windows, and each window was assigned with a class beat or no-beat, using the ECG as the gold standard. To detect the R peak in the ECG, the Pan–Tompkins algorithm [39] was used.

The sliding-window algorithm was implemented on the LDV signal, as shown in Figure 3. The algorithm used windows made of a defined number ($W_{d}$) of samples. The $W_{d}$-sized window shifted by a defined number ($W_{s}$) of samples, until it exceeded a newly-detected R peak in the ECG by a predefined number ($W_{overlap}$) of samples. This process is repeated for each identified R peak. It is worth noticing that the ECG was here used for data annotation purposes only. Each $W_{d}$-sized window starting in correspondence of the R peak in the ECG was labeled as *beat*. All the other windows among two consecutive R peaks were labeled as *no-beat*.

To define the size of $W_{d}$, $W_{s}$ and $W_{overlap}$, an analysis of the acquired LDV signals was conducted. In Figure 4, each curve shows the average of the 28 (800 ms-long) LDV signals, where each signal starts from an R peak in the ECG. The red circle highlights the mechanical effect of the pulsewave on the carotid wall captured by LDV. Hence, in the LDV signal, the blood pulse induced by the heartbeat has a time delay with respect to the QRS complex of the ECG [40]. The time delay was equal to 102 ±15 ms while the mean peak width was 142 ± 19 ms (this is in accordance with the literature [40]). Guided by these considerations, we hypothesized that the pulsewave can be found in a window $W_{d}$ = 75 samples (corresponding to 150 milliseconds) soon after the R peak. $W_{s}$ = 15 samples and $W_{overlap}$ = 60 samples were experimentally set to allow an evenly distributed selection of the entire LDV signal.

At the end of the annotation procedure the ratio between the annotated *no-beat* samples and the *beat* samples was about 10. Considering this class imbalance, to avoid our ML classifiers learning only the majority class, we decided to undersample the majority class by randomly selecting for each *beat* window a *no-beat* window. This procedure is usually conducted in the literature [41]. This undersampling resulted in 55,274 windows, equally balanced between the *beat* and *no-beat* class consisting of the LDV-beat dataset.

### 2.3. LDV-Signal Classification

Previous literature [37,38] suggests that Support Vector Machine (SVM) is one of the most valuable methods in ECG signal analysis. Besides SVM, also Decision Trees (DT), k-Nearest Neighbors (KNN) and Random Forest (RF) have been proven to be valuable classifiers for automatic recognition of arrhythmia from ECG [42,43,44]. Inspired by the literature, in this study we decided to investigate the four classifiers for LDV signal analysis. Specifically, each LDV-signal window was given as input to the four ML classifiers. Each classifier gave as output a class (beat or no-beat) for that window (i.e., a prediction for that window to contain a beat or not). It is worth noticing that no feature extraction was performed here, and the raw LDV signal windows were processed by the classifiers.

To perform LDV signal-window classification in *beat* and *no-beat*, we considered SVM with the Gaussian kernel (Ψ) [45]. Indeed, SVM allowed overcoming the *curse-of-dimensionality* that arises from analyzing our high-dimensional feature space [46,47]. The *kernel-trick* prevented parameter proliferation, lowering computational complexity and limiting over-fitting. Moreover, as the SVM classifications are only determined by the support vectors, SVMs are robust to noise in training data.

For our binary classification problem, given a training set of *N* data ${\left\{y_{k},x_{k}\right\}}_{k=1}^{N}$, where $x_{k}$ is the $k^{th}$ input feature vector and $y_{k}$ is the $k^{th}$ output label, the SVM decision function took the form of:(1)$$f\left(x\right)=sign\left[\sum_{k=1}^{N}a_{k}^{*}y_{k}\Psi \left(x,x_{k}\right)+b\right]$$
where:(2)$$\Psi \left(x,x_{k}\right)=exp\{-\gamma ||x-x_{k}{\left|\right|}_{2}^{2}/\sigma^{2}\},\;\gamma >0$$
*b* is a real constant and $a_{k}^{*}$ is retrieved as follows:(3)$$a_{k}^{*}=\max\left\{-\frac{1}{2}\sum_{k,l=1}^{N}y_{k}y_{l}\Psi \left(x_{k},x_{l}\right)a_{k}a_{l}+\sum_{k=1}^{N}a_{k}\right\}$$
with:(4)$$\sum_{k=1}^{N}a_{k}y_{k}=0,\;0\leq a_{k}\leq C,\;k=1,\ldots ,N$$

Both DT and RF are ML methods based on classification trees. DT learning uses a decision tree as a predictive model, mapping signal features to the relevant target value [48]. In the tree, leaves represent class labels and branches represent conjunctions of features that lead to class prediction. The RF classifier is an ensemble method that uses bagging by combining the output of several weak classifiers [49]. The main idea of ensemble methods is that a high number of weak learners can be used to create a strong learner. For RF, DT is used as a weak learner. Each RF consists of many individual DTs, where each tree is a classifier by itself that is given a certain weight for its classification output. The final classification decision is taken by averaging the class assignment probabilities calculated by all produced trees. The RF classifier is computationally efficient and less prone to overfitting with respect to other classifiers (e.g., KNN) [50].

KNN is a non-parametric method highly adopted in classification tasks [51]. KNN is the one with the low-computation complexity. The main concept behind KNN classification is that data points of the same class should be closer in the feature space. Hence, an input data point is classified by a majority vote of its K neighbors, with the label being assigned to the most common class among the K nearest neighbors in the training set. K is a positive integer, which is typically odd and small. The distance among neighbors is computed using different methods, such as the Euclidean, Mahalanobis, Minkowski and Manhattan distance. The performance of this algorithm greatly depends on two factors: a suitable distance function and an appropriate value for K. If K is too large, large-sized classes will overwhelm the small-sized one.

In this study, the hyper-parameters for each classifier were retrieved via grid-search and 5-fold cross validation on the training set. The grid-search space for γ and *C* in SVM classification was set to: [0.001, 0.01, 0.1] and [1, 10, 100], respectively. For the RF, the grid-search space was set to [3, 4, 5] for the maximum tree depth, and [50, 100, 150, 200, 250, 300, 350, 400, 450, 500, 600] for the maximum number of trees. The maximum tree depth in the DT classifier was retrieved with a grid-search space of [5, 20] with four values spaced evenly. The grid values were empirically set looking at the performance on the validation set, i.e., the test set was excluded from hyper-parameter tuning. The number of neighbors for KNN was tuned in a space equal to [3, 5, 7, 9]. Low values of tree depth and number of neighbors were set to prevent overfitting.

To evaluate the performance of the ML classifiers on the test set, the leave-one-out testing scheme was used here. Hence, it provides an almost unbiased estimate of the generalization ability of a classifier, especially when working with small datasets as ours [52]. Each time, the LDV windows from one subject were used for testing the performance of the classifier trained with the windows of all the other subjects. The separation at the subject level was necessary to prevent the classifiers to overfit on subject-specific features.

We further tested the ML classification performance on the full-length (10 min long) LDV signals collected from the 28 subjects, to simulate a real-life application scenario. It is worth noticing that the classifiers were tested on the LDV signals only (i.e., the ECG was used only as the gold standard). The results of the classification were then compared with the synchronous ECG signal to evaluate the classification performance. For this experiment, we implemented the leave-one-out training procedure, too. Furthermore, to prove the feasibility in a real scenario, the result of the classification on the full-length (10 min long) LDV signals was used to compute the HR from the LDV carotid signals in comparison with the standard ECG.

The classification was performed using scikit-learn (http://scikit-learn.org) on a computer equipped with an Intel Core i7-2.6 GHz and 8 Gb of RAM.

### 2.4. Performance Metrics

Inspired by previous work in the literature, which adopts ML for cardiac signal analysis [37,53], to evaluate the classification performance we computed the area (AUC) under the receiver operating characteristic (ROC) curve and standard indicators based on the confusion matrix, i.e., the class-specific classification recall ($Rec_{i}$), precision ($Prec_{i}$), and f1-score ($f1_{i}$):(5)$$Rec_{i}=\frac{TP_{i}}{TP_{i}+FN_{i}}$$
(6)$$Prec_{i}=\frac{TP_{i}}{TP_{i}+FP_{i}}$$
(7)$$f1_{i}=\frac{2\times Prec_{i}\times Rec_{i}}{Prec_{i}+Rec_{i}}$$
where $TP_{i}$, $FN_{i}$, $FP_{i}$ are the correctly classified positive samples, the false negatives and false positives samples for the ith class, respectively.

The ANOVA test for repeated measures (significance level = 0.05) was used to assess whether the classification achieved with our best performing classifier significantly differs from the others.

## 3. Results

In Figure 5, the ROCs obtained with the tested classifiers are reported. The curve for each leave-one-out run is plotted along with the mean curve among all the 28 runs, for each of the tested classifiers. The obtained AUC reached a mean value of 0.96 (RF), 0.90 (DT), 0.94 (KNN) and 0.95 (SVM). No statistical differences were found between the AUCs obtained with the tested classifiers (*p*-value = 0.08). Two subjects showed a lower value of the AUC with respect to the other 26 subjects. This was seen for all the classifiers. The reason may be attributed to the peculiar characteristics of the LDV in those two subjects, as shown in Figure 6. The red LDV signals, which refer to the two subjects, had a different shape with respect to the other signals.

Figure 7 shows the boxplots of the Rec, Prec, f1 for the class *beat*. The SVM achieved better performance when compared with the other classifiers, with median values for f1, Rec and Prec of 0.96, 0.96, 0.96, respectively. Table 1 shows the median value of the Prec, Rec and f1 for the class *beat* and *no-beat*, for each classifier. Hence, each leave-one-out training run resulted in a value of Prec, Rec and f1, for a total of 28 values for each classifier. No statistical differences were found between the four models in terms of f1 (*p* = 0.98) and Prec (*p* = 0.98) for the *beat* class. Instead, we found statistical differences when comparing the Rec of the *beat* class of SVM with respect to that of RF and DT. The SVM correctly classified 22,374 samples, with only 5263 (one order of magnitude lower) misclassified signal windows. Here, we decided to show only the results relevant to SVM for space constraint.

In Figure 8, an example of the results obtained when testing the SVM on one full-length, (10 min long), LDV signal is shown. The test consists of classifying a portion of signal with a length of $W_{d}$ = 75 samples with a sliding windows approach presented in Section 2.2. In this case, $W_{s}$ = 15 samples were used to improve the computational capacity of the classification having a better simulation in real-time. The blue lines refer to the probability for a window to belong to the *beat* class, as resulted from the SVM classification. In each graph is reported the synchronous ECG signal with the R peaks highlighted by gray bands. The same procedure was performed on each LDV signal acquired on the 28 subjects. The macro-average metrics were used for performance evaluation in the full-length (10 min long) signals, because the signals presented highly imbalanced class distribution. As suggested in the literature [54], macro-average metrics allow to better represent the overall classifier performance in the presence of imbalanced data.

Macro performance was computed, obtaining a macro-*Rec*, macro-*Prec* and macro-*f1* median value of 0.76, 0.75 and 0.76, respectively, as shown in Figure 9. The subject who obtained the best results achieved a macro-*Rec*, macro-*Prec* and macro-*f1* of 0.87, 0.85 and 0.84, respectively. The results in terms of ROC curves with the mean AUC are shown in Figure 10, the mean AUC among the 28 signals of 0.83.

Figure 11 shows an example of the computed HR using the result from the SVM classifier on the full-length (10 min long) LDV signal in comparison with the HR obtained from the ECG for one subject. To compute the HR from the SVM classification, for this test, only the windows that show a heartbeat probability result higher than 0.9 were considered. Only the heartbeats with higher probability were considered in case of adjacent detected heartbeats.

The differences of the mean HRs resulting from the LDV analysis and the ECG for each subject were computed and shown in Figure 12. The median error of 7.8 bpm between the LDV and ECG was found.

## 4. Discussion

In this paper, we presented an innovative learning-based framework for heartbeat detection from carotid LDV signals. To investigate the research hypothesis that ML may detect heartbeats from LDV signal acquired from the carotid artery, we evaluated the performance of four ML algorithms (i.e., RF, DT, KNN and SVM) on the newly collected LDV-beat dataset. The dataset consists of 320 min of carotid LDV recordings collected from 28 subjects. The dataset also provides the corresponding ECG signals, used as the gold standard for beat detection.

All ML classifiers achieved real time testing performance. As shown in Figure 5, similar ROCs were obtained for all the tested classifiers. Two subjects presented lower values of the AUC for all classifiers, which may be attributed to peculiar characteristics of their LDV signals as shown in Figure 6. This issue may be attenuated by enlarging the training set to encode a larger variability in terms of LDV signal shapes. It is also possible to consider using an LDV signal in different physical conditions for the same subject (e.g., after a walk, after an intense sport session, while sleeping).

As shown in Section 3, SVM achieved the best f1. This may be attributed to the ability of the SVM of handling high-dimensional feature-space (which was high if compared with the number of subjects in the LDV-beat dataset) and its robustness of tackling the noise components of the LDV signal. Similar conclusions have already been drawn in closer research fields [55,56,57].

The probability peaks in Figure 8 were associated with the LDV signal windows that were right after the R peaks in the ECG, supporting the promising performance of SVM. This happened in the LDV windows corresponding to the T wave in the ECG. Hence, the two windows in the LDV (i.e., those corresponding to the R and T peaks in the ECG) shared a similar waveform. The similar waveform may be attributed to the contribution of the blood passage in the jugular vein [26]. Nonetheless, despite the probability being larger than 50%, it is worth noticing that the probability for the heartbeat window was always larger than 80%. This supports our hypothesis that ML can tackle the variability encoded in LDV signals.

To simulate a real application scenario, we tested the performance of SVM on the full-length (10 min long) LDV signals. The achieved performance was comparable with that obtained when testing the classifier on the LDV-beat dataset, showing that ML algorithms may be successfully exploited also with data acquired in real settings. When using the result of the classification to compute the HR from LDV signals, a difference less than 10 bpm with respect to the ECG analysis was achieved, showing a great potential in the application. It is worth noticing that the procedure to measure the HR from the results of the classifier must be improved. This may represent a preliminary proof that the proposed method could be used in real-life applications. It is worth noticing that, in our experiments, a subject’s movement caused by swallowing or breathing did not represent an issue for ML classifiers, probably because movement is usually characterized by a lower frequency content with respect to heartbeat [58]. Hence, subjects were left free to breathe and swallow normally and this did not seem to affect the classification performance.

A limit of the proposed framework may be seen in the processing of low-quality LDV signals. Hence, if the laser beam does not point in the optimal position over the carotid, the signal-to-noise ratio (SNR) may be too low to perform proper LDV signal processing. In this work, we manually excluded one subject for which the SNR was low. However, a real-time filter can be used to improve detection in noisy signals.

A further limit may be seen in the enrolment of subjects in the study. All subjects were healthy and reported to have a normal lifestyle. However, athletes, for example, may develop cardiovascular dynamics (e.g., bradycardia, athlete heart syndrome), which may affect the performance of the ML classifiers. Similar issues can be encountered on subjects affected by arrhythmia. For example, a premature ventricular contraction leads to a reduction of ventricular filling and decreases the peripheral pulse amplitude [59]. We indeed recognized that a limitation of the presented methodology could be seen in the relatively small number of subjects acquired to collect our dataset. Unfortunately, no publicly available datasets of LDV signals collected from subjects with pathological conditions are available today. As future work, to further prove the potentiality of ML in this field of research, a study will be conducted with an enlarged dataset. Nonetheless, it is worth noticing that this work is designed as a proof of concept to assess the potentiality of ML for LDV signal processing, paving the way for more advanced analysis and experiments in the future. Enlarging the dataset (in terms of size and variability) will allow other researchers in the field to test more advanced ML algorithms and to design other clinically relevant applications.

As future work we would like to investigate deep learning-based strategies, which are today the state of the art in the field of medical-image analysis [60,61,62]. Hence, preliminary researches on deep learning algorithms for ECG analysis are providing promising results, but further investigation is needed in the LDV field [63,64,65]. Deep learning-based model, able to process temporal information naturally encoded in signals, can be investigated. Inspired by [66], the analysis of spatio-temporal features to investigate arterial stiffness from carotic LDV signals, may represent a challenge with promising clinical application.

## 5. Conclusions

In this paper, we presented the first application of ML algorithms to the problem of heartbeat detection from LDV carotid signals. LDV may not be suitable for long-term monitoring due to its strict correlation to the measurement point, but it may be a valuable alternative to ECG for specific cases where non-contact measurements are required. Examples include cases in which infection hazard is a major issue and when magnetic-resonance scanning has to be performed. The results achieved on the LDV-beat dataset (the first publicly available annotated dataset in the field), suggested that ML can be successfully exploited to perform heartbeat detection on a carotid LDV signal. By releasing our LDV-beat dataset to the scientific community, we hope to stimulate researchers to translate the use of ML for LDV signal analysis, with a view to extend the proposed work and move it to clinical practice.

## Figures and Tables

**Figure 1 sensors-20-05362-f001:**
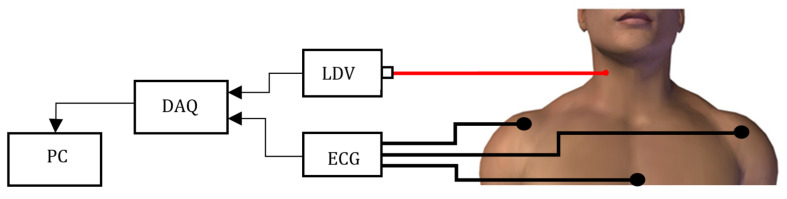
The LDV measurement set-up is composed by a Laser Doppler Vibrometer (LDV) pointed on the subject’s skin over the carotid. The LDV signal is generated and fed to a PC by a DAQ board.

**Figure 2 sensors-20-05362-f002:**
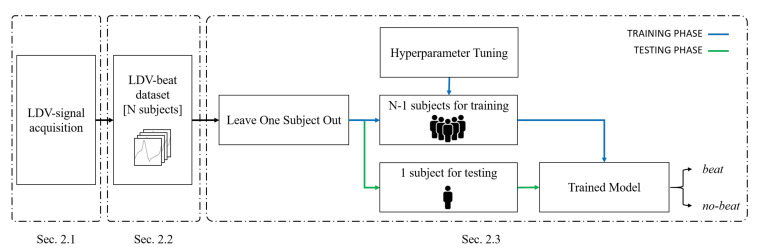
Workflow of the proposed machine-learning (ML) framework to heartbeat detection from carotid LDV signal.

**Figure 3 sensors-20-05362-f003:**
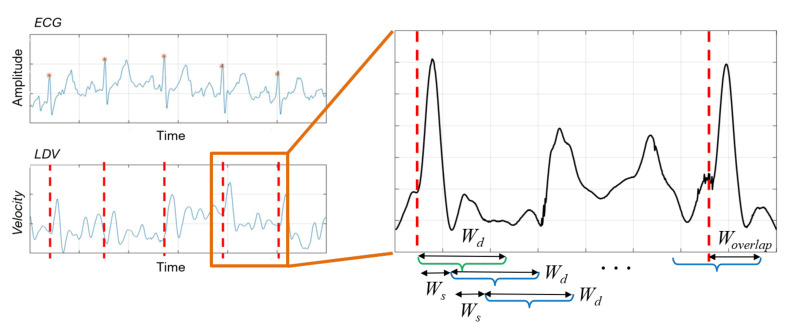
Sliding-window algorithm used to extract LDV signal windows. $W_{d}$ window size, $W_{s}$ =shift length, $W_{overlap}$ = overlap with a new beat. The green and blue windows denote *beat* and *no-beat* samples, respectively. (The LDV signal was smoothed for visualization purposes only).

**Figure 4 sensors-20-05362-f004:**
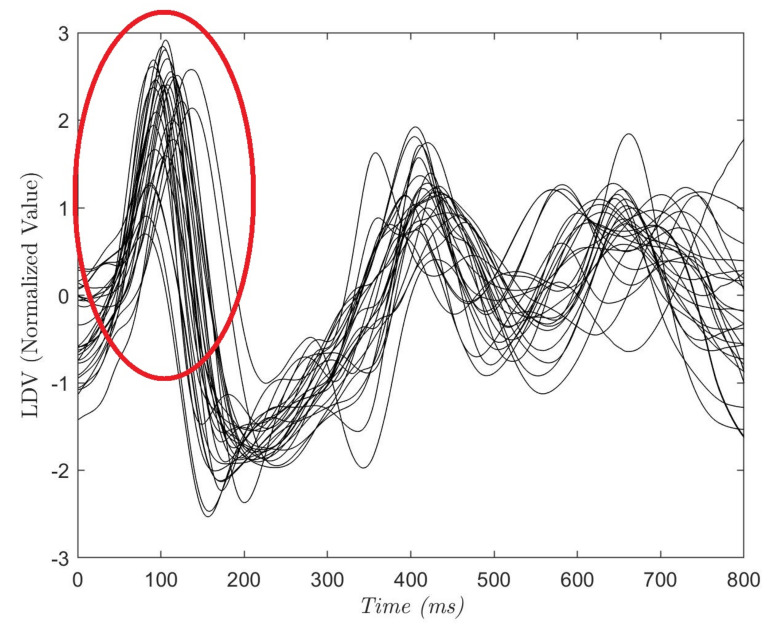
Each curve refers to one subject and is the average of a portion of the LDV signal. The portion starts from the R peak in the electrocardiograph (ECG) and last 800 ms. The red circle shows the LDV pulse wave peak that can be found at 102 ± 15 ms after the R-peak.

**Figure 5 sensors-20-05362-f005:**
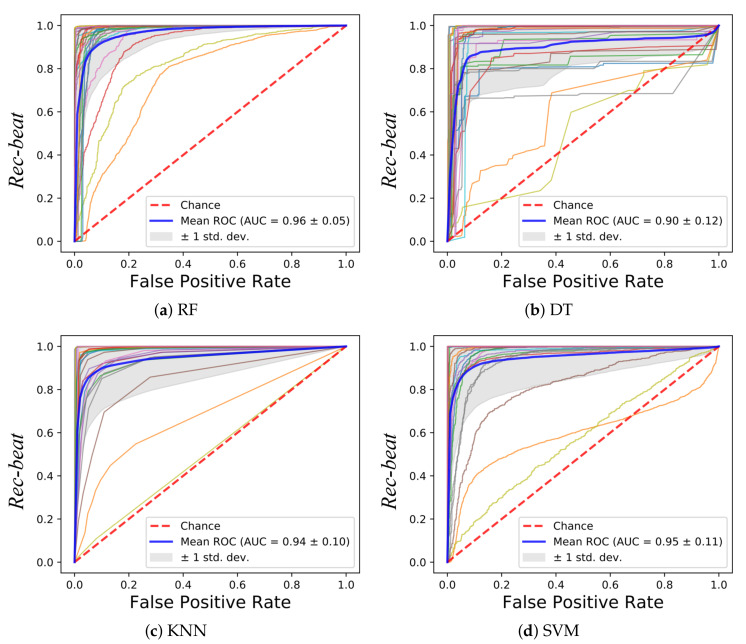
Receiver operating characteristic (ROC) curves obtained using the LDV-beat dataset with (**a**) Random Forest (RF), (**b**) Decision Tree (DT), (**c**), K-Nearest Neighbor (KNN) and (**d**) Support Vector Machines (SVM). Each ROC refers to a subject. The area (AUC) under the ROC is shown, too.

**Figure 6 sensors-20-05362-f006:**
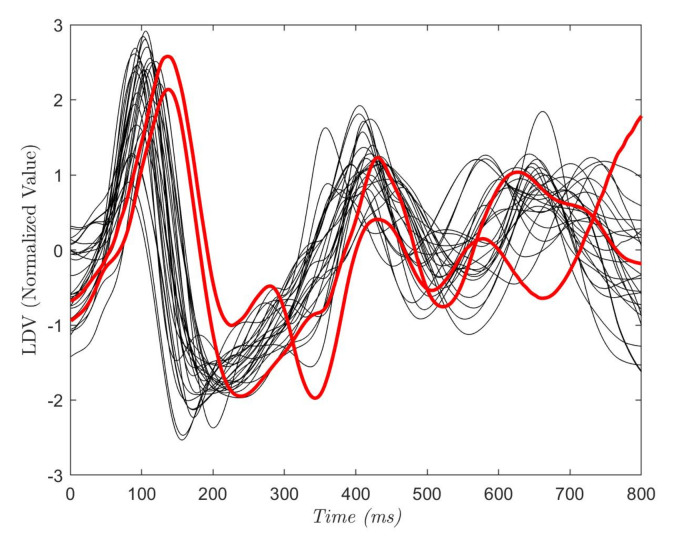
Each curve refers to one subject and is the average of a portion of the LDV signal. The portion starts from the R peak in the ECG and last 800 ms. The red curves refer to the subjects with the lowest AUC values.

**Figure 7 sensors-20-05362-f007:**
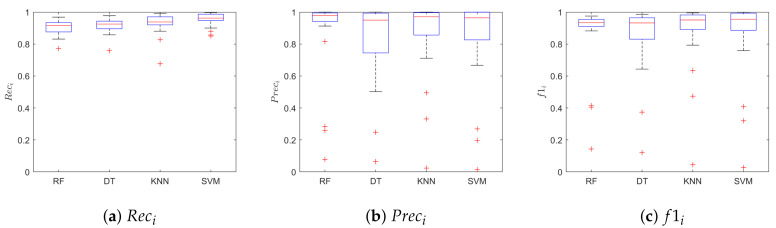
Classification results obtained on the LDV-beat dataset. Results are shown in terms of: (**a**) recal (*Rec_i_*), (**b**) precision (*Prec_i_*), and (**c**) f1-score (*f*1*_i_*) for the class *i = beat*. The results are obtained when using Random Forest (RF), Decision Tree (DT), K-Nearest Neighbor and Support Vector Machines (SVM).

**Figure 8 sensors-20-05362-f008:**
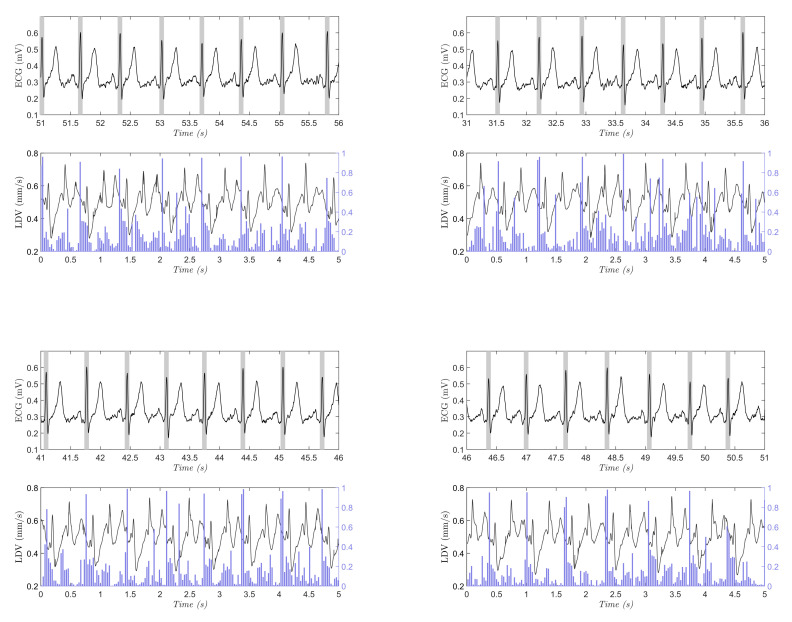
Four sample testing results on the full-length (10-min long), LDV signal (the signal was smoothed for visualization purposes only). In each subfigure, the ECG signal with gold-standard beats (grey rectangles centered with the R peak from the ECG), and the output probability (vertical blue line) for the *beat* class obtained with support vector machines are shown. The beats in the ECG signal are shown with rectangles for better visibility.

**Figure 9 sensors-20-05362-f009:**
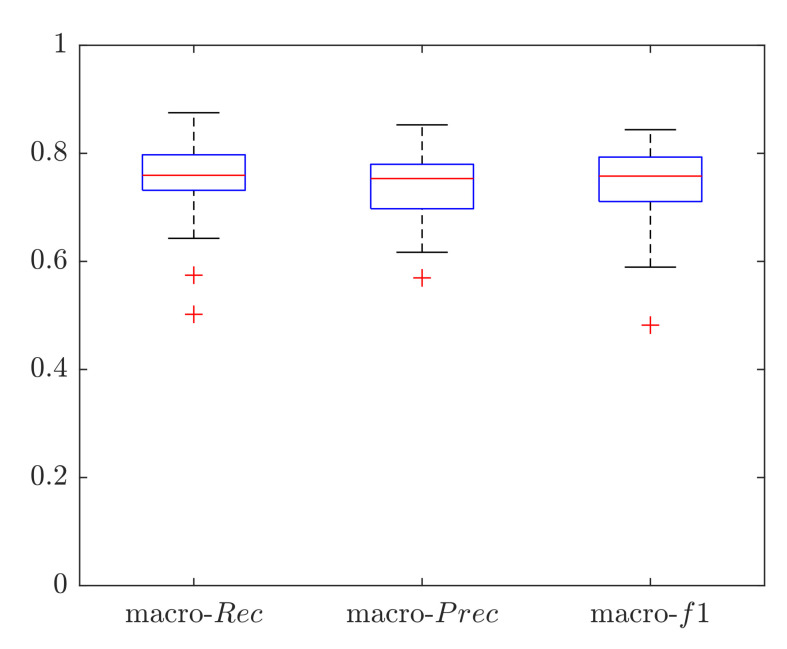
Classification results in terms of macro-average metrics obtained when classifying the full length (10-min long) LDV signals using a Support Vector Machine (SVM). Results are shown in terms of: macro recall (macro-Rec), macro precision (macro-Prec), and macro f1-score (macro-f1).

**Figure 10 sensors-20-05362-f010:**
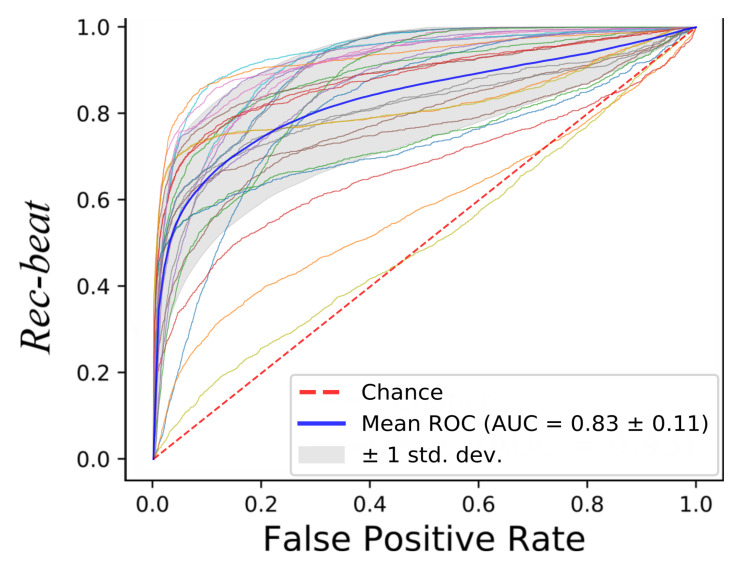
Receiver operating characteristic (ROC) curves obtained when classifying the full-length (10-min long) LDV signals with Support Vector Machines (SVM).

**Figure 11 sensors-20-05362-f011:**
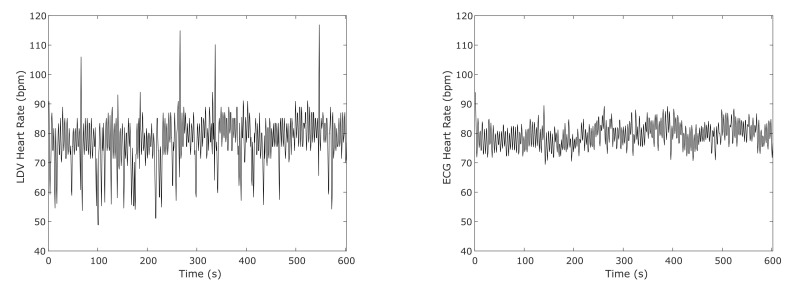
An example of the heart-rate (HR) computed from an SVM classifier on the full-length (10 min long) LDV signal and from ECG in one subject.

**Figure 12 sensors-20-05362-f012:**
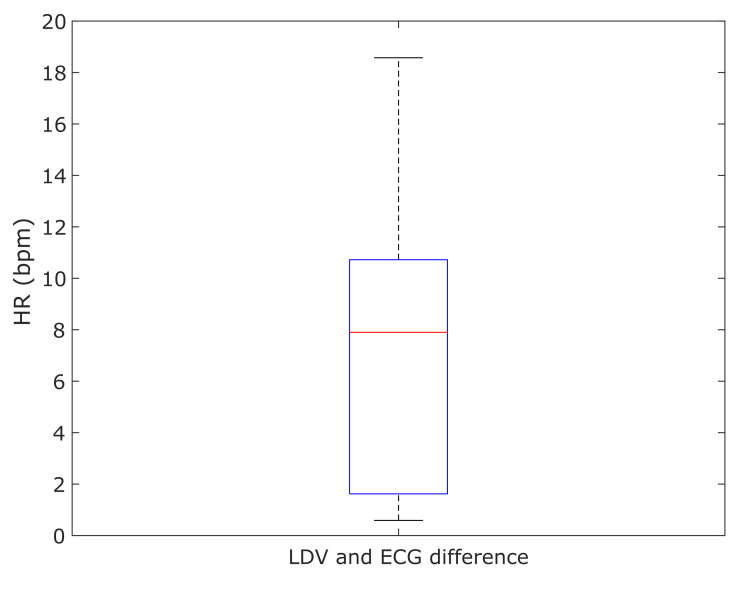
Boxplot of the absolute differences between the mean value of the heart rate (HR) computed with the LDV and the ECG in the 28 subjects.

**Table 1 sensors-20-05362-t001:** Classification performance obtained when classifying the LDV-beat dataset with Random Forest (RF), Decision Tree (DT), K-Nearest Neighbors (KNN) and Support Vector Machine (SVM) classifiers. Classification recall ($Rec_{i}$), precision ($Prec_{i}$), and f1-score ($f1_{i}$) are reported for the class *beat* (*i* = 1) and *no-beat* (i = 0).

	*Prec*		*Rec*		*f*1	
	*Beat*	*No-Beat*	*Beat*	*No-Beat*	*Beat*	*No-Beat*
*RF*	0.98	0.93	0.92	0.98	0.93	0.93
*DT*	0.95	0.94	0.93	0.96	0.93	0.94
*KNN*	0.97	0.95	0.94	0.98	0.95	0.95
*SVM*	0.96	0.97	0.96	0.96	0.96	0.98

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
