# Peer review of "Heartbeat Detection by Laser Doppler Vibrometry and Machine Learning"

_sensors, 2020, doi:10.3390/s20185362_

Round 1

Reviewer 1 Report

This manuscript proposed to detect heartbeat from velocity signal measured by Laser Doppler Vibrometer (LDV). The authors listed the advantages of this approach (e.g., non-contact, MRI compatible, etc) and claimed that it was a promising technique for monitoring of ECG. However, the laser beam has to point properly over the carotid artery of the subject. As described in the authors’ study, the subjects were lying in supine position to limit involuntary movements. Even under this strict condition, the measurement of one subject was excluded from 29 subjects “because the laser was not pointed properly over the carotid artery”. Therefore, LDV approach may not be feasible for heart monitoring, especially for long-term monitoring.

In this study, a sliding-window algorithm was used to divide the LDV signal into segments and then tried to classify each segment as beat or no-beat using several machine learning algorithms. There are several concerns: (1) The location of R peak in the ECG signal was required to divide the LDV signal into overlapping windows, shown in Figure 3, but ECG signal would not be available in real application; (2) The window parameters (Wd, Ws and Woverlap) are all determined based on healthy subjects. In the result, two subjects showed a lower value of the AUC. The authors thought it might be caused by the different shape of their LDV signals. What will be the shape of the LDV signal for patients with cardiovascular disease and how to deal with this problem?

In Section 2.3, it roughly mentioned the basic concept of the four machine learning algorithms, but what the input feature vector was is not clearly described. Is it the raw signal or extracted features? In this study, the heartbeat detection problem is simplified as a binary classification problem. However, the location of the LDV pulse wave peak is also important. The result on the 1-min LDV signal is interesting, but how this result is obtained is not clear and Figures 8 and 9 are confusing. In Figure 8, why do the blue peaks match the ECG R wave well? Is it the first window after each R wave? In “From one window classified and the following window, a 15 point step was used to obtain real time performances.”, what does “the 15 point step” mean?

In addition, please double check the references. For example, [67] is CVD Statistics, not “ECG-based applications”. [61-63] are all deep learning for image processing, probably [64-66] are in closer fields.

Last suggestion is to correct the language and the grammar mistakes and polish the manuscript.

Reviewer 2 Report

The paper presents a study on applicability of laser vibrometry to non-contact estimation of heartbeat. The approach is interesting and the applicability is evident, but there are several issues that should be considered before the paper is suitable for publication:

i) In laser vibrometry we are facing several sources of noise which (more or less) influences the measurements. Usually various filters are applied to reduce the noise (especially the frequencies we are not interested in). The paper does not address the LDV signal filtering but from figures shown we can assume that some filtering (smoothing) was done. The way the acquisition was done together with details on filtering should be explained.

ii) It is not clear what is the output of the analysis. Beat is simply to general information. The output data should be more precisely described.

iii) I believe that the windowing based on ECG signal is biased and should be avoided. It is suggested to use a standard fast Fourier transform instead to estimate the dominant frequency.

iv) There are several problems regarding the proposed methodology that should be more properly addressed and discussed in the paper. Machine learning results are based on the data obtained on healthy young subjects. Are they of any value for a person with irregularities or of different age group? The motivation for using machine learning instead of more standard methods for analyzing time series  should also be more firmly stated.

Round 2

Reviewer 1 Report

First, the reason of forming the heartbeat detection problem as a binary classification problem is not quite clear although this is the main part of the proposed work. What is the medical significance of detecting beat or no-beat for each time window? Second, as I mentioned previously, the result on the 1-min LDV signal is more interesting. However, in this manuscript, only one-minute result was provided as an example. In this study, 10-minutes data were recorded for each subject, why not completing analysis for all of them? From Figure 8, it seems that the output peaks match the R peak pretty well.

Grammar mistakes still exist, e.g., Line 240: “two subject”, Line 328: “LDV may be not be suitable”.

Round 3

Reviewer 1 Report

I got confused with the new results. What are the differences between the new results and the previous results using SVM? Could more detail be provided? Does the same window apply to the 10-minute data? Still Wd=75 and Ws=15? What is the difference between Fig. 9 and the SVM result in Fig.7, as well as Fig. 10 and Fig. 5(d)? Why the new results are worse?

I think I did not make it clear last time. From Fig. 8, we can see that the highest output probability matched the R peak well. Could the R peaks be detected from these outputs, and heart rate be calculated? If so, the calculated heart rates from the LDV signals can be compared with those detected by the ECG. I do hope to see heart rate instead of beat or no-beat window.

Grammar mistakes still exist, e.g., Line 248: still “two subject”; Line 54: double period; Line 274: no period.
